# EGO TO WORLD: COLLABORATIVE SPATIAL REASONING IN EMBODIED SYSTEMS VIA REINFORCEMENT LEARNING

## ABSTRACT

Understanding the world from distributed, partial viewpoints is a fundamental challenge for embodied multi-agent systems. Each agent perceives the environment through an ego-centric view that is often limited by occlusion and ambiguity. To study this problem, we introduce the Ego-to-World (E2W) benchmark, which evaluates vision–language model's ability to fuse heterogeneous viewpoints across three tasks: (i) global counting, (ii) relational location reasoning, and (iii) action-oriented grasping that requires predicting view-specific image coordinates. To address this setting, we propose CoRL, a two-stage framework that combines Chain-of-Thought supervised fine-tuning with reinforcement learning using Group-Relative Policy Optimization. Its core component, the Cross-View Spatial Reward (CVSR), provides dense task-aligned feedback by linking reasoning steps to visual evidence, ensuring coherent cross-view entity resolution, and guiding the model toward correct final predictions. Experiments on E2W show that CoRL consistently surpasses strong proprietary and open-source baselines on both reasoning and perception-grounding metrics, while ablations further confirm the necessity of each CVSR component. Beyond that, CoRL generalizes to external spatial reasoning benchmarks and enables effective real-world multi-robot manipulation with calibrated multi-camera rigs, demonstrating cross-view localization and successful grasp-and-place execution. Together, E2W and CoRL provide a principled foundation for learning world-centric scene understanding from distributed, ego-centric observations, advancing collaborative embodied AI. Code is available at CoRL .

## 1 INTRODUCTION

Recent advances in Vision–Language Models (VLMs) have catalyzed significant progress in embodied intelligence. By grounding natural language within visual perception, VLMs have enabled a diverse range of embodied tasks, from instruction following to interactive manipulation. This progress has spurred extensive research into VLM applications across domains such as robotics (Kang et al., 2025), navigation (Wang et al., 2025b), and spatial reasoning (Zhou et al., 2025; Yin et al., 2025). Nevertheless, the majority of existing methodologies are confined to single-view scenarios, where perception is limited to an ego-centric or a fixed global viewpoint. Such a constraint inherently leads to incomplete scene understanding and restricted reasoning capabilities.

In many real-world applications, multiple heterogeneous agents—such as cooperative service robots in domestic environments or Vehicle-to-Everything (V2X) systems in autonomous driving—operate concurrently. In these settings, multi-agent coordination is not merely beneficial but essential. A solitary viewpoint is fundamentally susceptible to occlusions and partial observations, whereas integrating complementary perspectives from multiple agents can provide richer contextual understanding. As illustrated in Figure 1, cross-view compositional reasoning empowers agents to surmount these limitations and execute spatially grounded actions with high fidelity. However, achieving such reasoning capabilities is profoundly challenging, as it necessitates the integration of heterogeneous viewpoints, the resolution of cross-view ambiguities, and the alignment of overlapping observations to construct a coherent scene representation.

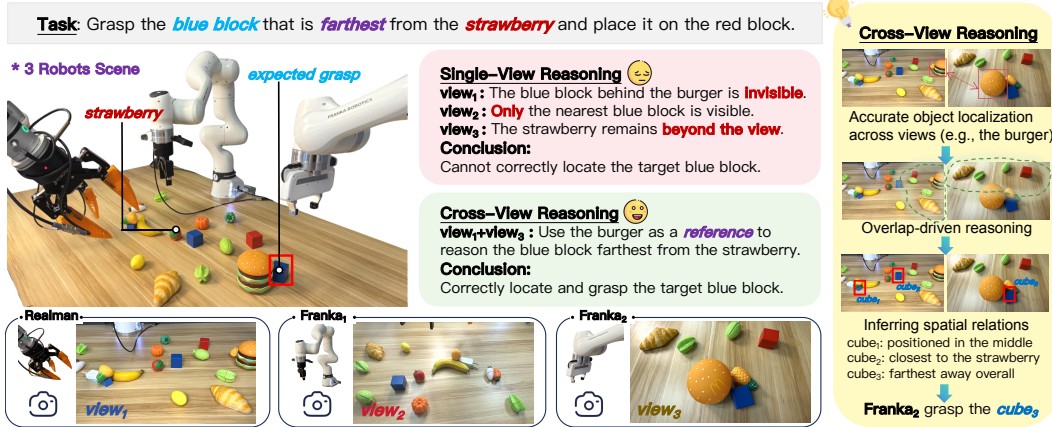

Figure 1: An illustration of collaborative spatial reasoning in embodied systems. Reasoning from a single viewpoint fails due to occlusions or a limited field of view. In contrast, cross-view compositional reasoning integrates multiple perspectives to correctly localize and grasp the target object—the blue block farthest from the strawberry.

To bridge this critical gap, we investigate the problem of collaborative spatial reasoning in multi-agent embodied systems. We formalize a novel setting wherein each agent contributes its partial, ego-centric observations, and the VLM need to integrate these disparate inputs to perform complex spatial reasoning and perception tasks. To this end, we introduce the **Co**llaborative Spatial **R**easoning Reinforcement **L**earning (CoRL) framework, which is augmented by a novel **C**ross-**V**iew **S**patial **R**eward (CVSR). The CVSR is meticulously designed to explicitly incentivize policies that: (i) consistently localize objects across different viewpoints, (ii) effectively fuse complementary ego-centric information, and (iii) maximize collective scene coverage. By shaping the learning signal around these core principles of spatial grounding, CVSR compels the VLM to transcend single-view perception and develop robust collaborative spatial reasoning abilities. Through systematic comparisons of alternative training paradigms, we demonstrate that our proposed Supervised Fine-Tuning (SFT) followed by Reinforcement Learning (RL) pipeline, empowered by CVSR, consistently achieves state-of-the-art performance on collaborative perception tasks.

In summary, our contributions are threefold:

◇ We pioneer the formalization of collaborative perception and reasoning with VLMs in multi-agent embodied systems, and introduce a large-scale dataset specifically tailored for this setting.

◇ We propose the CoRL framework, featuring the novel Cross-View Spatial Reward (CVSR), which explicitly incentivizes the fusion of ego-centric views and enhances spatial grounding.

◇ We empirically validate the effectiveness and generalizability of our approach across diverse VLMs, showing substantial improvements over strong baselines and robust transfer to external spatial reasoning benchmarks.

## 2 RELATED WORK

**Embodied Multi-Agent Cooperation.** Coordination among multiple agents is a fundamental challenge in embodied AI. Research in this domain has historically focused on high-level task allocation Obata et al. (2024); Wang et al. (2024); Liu et al. (2025a) and joint decision-making Zhang et al. (2023a); Wang et al. (2025a). More recently, the advent of Large Language Models (LLMs) has catalyzed progress in multi-agent collaboration, enabling sophisticated distributed planning and communication strategies Bo et al. (2024); Guo et al. (2024); Nasiriany et al. (2024); Zhou et al. (2023). However, a primary limitation of these LLM-based approaches is their reliance on symbolic or textual representations, which are detached from the visual world. This detachment restricts their capacity to handle perceptual ambiguities and perform fine-grained spatial reasoning. While a few pioneering studies have started to integrate Vision–Language Models (VLMs) into multi-agent

systems Wang et al. (2025b); Zhang et al. (2024a); Kang et al. (2025), they typically treat each agent's viewpoint in isolation or default to single-view reasoning. In stark contrast, our work directly confronts this limitation by proposing a framework centered on compositional, cross-view reasoning to achieve deeper and more effective collaboration.

**Spatial Understanding.** Spatial understanding—the ability to parse intricate geometric configurations, spatial layouts, and object interrelations from diverse visual inputs—is critical for intelligent systems across a wide range of domains, from geometric problem-solving Gao et al. (2023); Shi et al. (2024); Zhang et al. (2024b) to embodied robotics Hu et al. (2023); Ji et al. (2025). To bolster the spatial reasoning capabilities of VLMs, recent works have explored increasingly advanced training methodologies. Techniques such as multi-stage supervision with Chain-of-Thought (CoT) prompting Xu et al. (2024); Wei et al. (2022) and reinforcement learning (RL) with carefully engineered reward mechanisms Guo et al. (2025) have yielded significant gains. Subsequent research has further underscored the importance of highly tailored reward designs for complex visual reasoning tasks Liu et al. (2025b;c); Tan et al. (2025). Building upon this line of inquiry, our work addresses the distinct and even more complex challenge of *multi-agent* spatial understanding. Here, the central problem is to synthesize a globally coherent and semantically consistent scene representation from fragmented, ego-centric observations. This necessitates not only unifying disparate multi-view visual data but also enforcing strong and robust spatial consistency across multiple, partially overlapping perspectives.

**Reinforcement Learning for Visual Reasoning.** Reinforcement Learning (RL) has emerged as a powerful and versatile paradigm for training intelligent agents that reason from high-dimensional visual data. Moving beyond the static nature of supervised pretraining, RL enables models to learn through direct and interactive environmental engagement, optimizing their policies via reward-driven feedback Liu et al. (2025b); Tan et al. (2025); Sarch et al. (2025); Chen et al. (2025). Foundational applications in perception-driven tasks, such as navigation Zhu et al. (2017) and manipulation Zhou et al. (2025); Kang et al. (2025), have demonstrated RL's strong efficacy in tightly coupling perception and action. More recently, the field has progressed toward structured reward designs for multi-modal reasoning, incorporating mechanisms like CoT guidance Zhang et al. (2024c) or geometric consistency constraints Jiang & Lu (2024) to further enhance embodied decision-making performance. Despite these advances, the predominant focus of RL-based visual reasoning has remained on single-agent, single-image settings, leaving multi-view reasoning largely underexplored. Our framework extends this paradigm to the multi-agent, multi-view context, introducing a novel and unified reward structure designed explicitly to foster collaborative spatial reasoning.

## 3 EGO-TO-WORLD TASK

To systematically evaluate collaborative spatial reasoning, we introduce the *Ego-to-World Benchmark* (**E2W-Bench**), which operationalizes a multi-agent, multi-view paradigm. As shown in Figure 2, multiple robotic agents capture partial ego-centric observations of a shared 3D environment. The central challenge for a Vision–Language Model is to integrate these fragmented perspectives into a coherent global scene representation and to answer natural-language queries or perform action-oriented predictions. E2W-Bench consists of two categories of tasks: spatial reasoning QA (E2W-1, E2W-2) and perception for grasping (E2W-3). Further dataset details are provided in Appendix C.

**E2W-1 (Counting).** This task evaluates the ability to aggregate object instances across overlapping views and output an accurate global count.

**E2W-2 (Location Reasoning).** Here the model must infer spatial relations among objects that never co-occur in a single view, requiring cross-view reasoning to answer correctly in natural language.

**E2W-3 (Grasping).** Unlike the QA tasks, E2W-3 requires action-oriented predictions. The model must translate a language command involving spatial relations into precise 2D coordinates within specific agent's viewpoint, thereby linking compositional reasoning to downstream robotic manipulation.

Together, these tasks provide a comprehensive benchmark that jointly evaluates high-level symbolic reasoning, fine-grained visual grounding, and embodied spatial referring across diverse scenarios. By explicitly linking abstract reasoning to actionable predictions, E2W-Bench offers a rigorous testbed for multi-view understanding and real-world transfer.

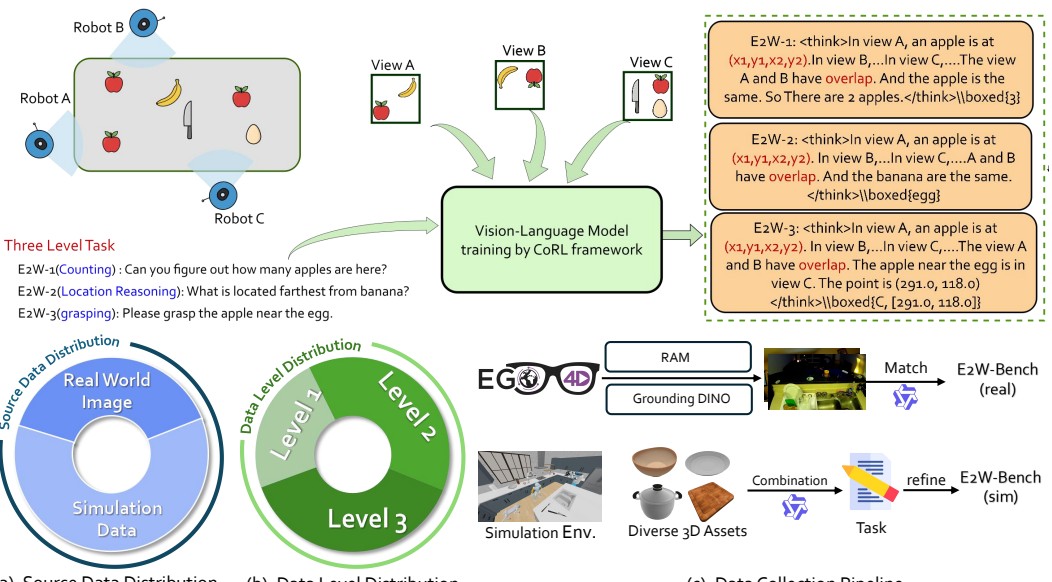

Figure 2: **Overview of the Ego-to-World (E2W) Benchmark. Top:** Multiple agents (Robot A, B, C) each provide partial ego-centric views of a shared scene. The vision language model trained with our CoRL framework integrate these complementary perspectives to solve three tasks: Counting (E2W-1), Location Reasoning (E2W-2), and Grasping (E2W-3). **Bottom:** The benchmark combines diverse real and simulated data and organizes them into varying complexity levels.

## 4 METHODS

### 4.1 OVERVIEW

We present ***CoRL*** (Collaborative Spatial Reasoning Reinforcement Learning), a framework that equips Vision–Language Models (VLMs) with collaborative perception and reasoning capabilities in multi-agent embodied settings. A central VLM aggregates and reasons over partial, ego-centric observations streamed by multiple agents, producing task-specific outputs under a unified interface. The overall architecture is shown in Figure 3.

### 4.2 PROBLEM FORMULATION

Consider a shared environment $\mathcal{E}$ populated by $N$ embodied agents $\mathcal{A} = \{a_1, \ldots, a_N\}$. At a given time, each agent $a_i$ observes an ego-centric RGB image $I_i \in \mathbb{R}^{H \times W \times 3}$. All agents receive a common natural-language query $Q$. A central VLM implements a policy $\pi_\theta$ parameterized by $\theta$ that consumes the multi-view input and the query to produce a prediction

$$\hat{y} = \pi_\theta\big(\{I_i\}_{i=1}^N, Q\big), \tag{1}$$

where $\hat{y}$ is either a textual response (for counting and relational reasoning) or a pair of image-plane coordinates (for grasping).

Training is cast as maximizing the expected task reward over a dataset $\mathcal{D}$ of instances $(\{I_i\}, Q, y)$ with ground-truth label $y$:

$$\max_\theta \ \mathbb{E}_{(\{I_i\}, Q, y) \sim \mathcal{D}}\big[ R(\hat{y}, y) \big]. \tag{2}$$

The reward function $R$ evaluates the quality of $\hat{y}$ against $y$ and is central to our method; its design is detailed in Section 4.4.

### 4.3 CoRL TRAINING PIPELINE

CoRL is trained in two stages: supervised fine-tuning for initialization, followed by reinforcement learning for policy refinement.

Figure 3: **CoRL framework.** The model is first initialized via supervised fine-tuning (SFT) on Chain-of-Thought annotations, then refined with reinforcement learning (RL). During RL, the policy is optimized with an format reward and the Cross-View Spatial Reward (CVSR), which supplies dense feedback on cross-view fusion and spatial consistency, guiding robust collaborative reasoning.

**Supervised Fine-Tuning.** We initialize $\pi_\theta$ with SFT on CoT-augmented data. Each training tuple is $(x, q, r, a)$, where $x$ denotes the multi-view inputs, $q$ is the query, $r$ is the intermediate reasoning trace, and $a$ is the final answer. Let $y = [r, a]$ be the concatenated target sequence. The SFT objective is the usual next-token log-likelihood:

$$\mathcal{L}_{\text{SFT}} = -\mathbb{E}_{(x,q,r,a)\sim\mathcal{D}} \sum_{t=1}^{|y|} \log \pi_\theta\big(y_t \mid x, q, y_{<t}\big). \tag{3}$$

This yields an initial policy $\pi_{\text{SFT}}$ (denoted $\pi_0$) that provides a strong starting point for RL.

**Reinforcement Learning Fine-Tuning.** After SFT initialization, we further optimize the policy with *Group Relative Policy Optimization* (GRPO). GRPO improves stability and sample efficiency by normalizing rewards within each sampled group of candidate responses, thereby computing *group-relative advantages* that reduce variance and sharpen credit assignment.

Concretely, for an input $u = (x, q)$, we draw $G$ candidate responses $\{y_j\}_{j=1}^G$ from the current policy $\pi_\theta$ and score each with a reward $R_j$ (Section 4.4). GRPO computes the empirical mean $\bar{R}$ and standard deviation $\sigma_R$ of the rewards and defines a standardized advantage for each candidate:

$$\mathcal{A}_j = \frac{R_j - \bar{R}}{\sigma_R}. \tag{4}$$

This group-relative normalization emphasizes responses that outperform their peers, rather than absolute reward magnitude, and thus stabilizes training.

To update the policy, we first define the probability ratio between the current policy $\pi_\theta$ and the reference policy $\pi_0$ from SFT:

$$r_j(\theta) = \frac{\pi_\theta(y_j \mid u)}{\pi_0(y_j \mid u)}. \tag{5}$$

The core of the GRPO objective is a clipped surrogate function, which constrains the policy update step size. This objective is defined as:

$$\mathcal{L}^{\text{CLIP}}(\theta) = \mathbb{E}_u \left[ \sum_{j=1}^G \min\Big(r_j(\theta)\mathcal{A}_j, \ \text{clip}\big(r_j(\theta), 1-\epsilon, 1+\epsilon\big)\mathcal{A}_j\Big) \right], \tag{6}$$

where $\epsilon$ is a hyperparameter that defines the clipping range.

The final policy parameters are updated by maximizing the full GRPO objective, which incorporates a KL divergence penalty to further regulate the policy update:

$$\mathcal{J}(\theta) = \mathcal{L}^{\text{CLIP}}(\theta) - \beta \, \text{D}_{\text{KL}}\big(\pi_\theta(\cdot \mid u) \, \| \, \pi_0(\cdot \mid u)\big), \tag{7}$$

where $\pi_0$ is the SFT-initialized reference policy and $\beta > 0$ regulates the trust region enforced by the KL regularizer. This formulation allows CoRL to directly optimize spatial reasoning rewards while preserving stability and sample efficiency, fully leveraging the group-relative advantage mechanism within a robust PPO-style optimization framework.

## 4.4 CROSS-VIEW SPATIAL REWARD (CVSR) DESIGN

The total reward combines an output-format component with a cross-view spatial component:

$$R = \lambda_1\, R_{\text{format}} + \lambda_2\, R_{\text{CVSR}}, \qquad \lambda_1, \lambda_2 > 0. \tag{8}$$

**Output-Format Reward** $R_{\text{format}}$. To ensure interpretability and reliable parsing, the model receives a binary reward for structural correctness. Specifically, the intermediate reasoning must be enclosed in `<think>...</think>` tags and the final answer must appear in a designated box; success yields $R_{\text{format}} = 1$, otherwise $0$. This encourages the model to articulate a reasoning trace prior to committing to an answer.

**Cross-View Spatial Reward** $R_{\text{CVSR}}$. CVSR delivers dense feedback targeted at collaborative spatial reasoning. It aggregates three components:

$$R_{\text{CVSR}} = w_{\text{ground}}\, R_{\text{ground}} + w_{\text{overlap}}\, R_{\text{overlap}} + w_{\text{ans}}\, R_{\text{ans}}, \tag{9}$$

with nonnegative weights $w_{\text{ground}}, w_{\text{overlap}}, w_{\text{ans}}$.

*(i) Grounding reward $R_{\text{ground}}$.* To align reasoning with visual evidence, the model is prompted to emit bounding boxes for key objects referenced in its chain of thought. Let $\hat{B} = \{\hat{b}_i\}_{i=1}^m$ be predicted boxes and $B^* = \{b_j^*\}_{j=1}^n$ ground-truth boxes. We compute an optimal bipartite matching $\sigma$ via the Hungarian algorithm that maximizes total IoU, and define

$$R_{\text{ground}} = \frac{1}{|\sigma|} \sum_{i=1}^{|\sigma|} \text{IoU}\big(\hat{b}_i,\, b_{\sigma(i)}^*\big), \tag{10}$$

which provides a dense localization signal.

*(ii) Overlap accuracy $R_{\text{overlap}}$.* To incentivize cross-view entity resolution, the model must report the number of unique object instances that appear in more than one view, denoted $\hat{n}_{\text{overlap}}$. Comparing to the ground truth $n_{\text{overlap}}^*$ yields

$$R_{\text{overlap}} = \mathbb{I}\big[\hat{n}_{\text{overlap}} = n_{\text{overlap}}^*\big], \tag{11}$$

encouraging the model to distinguish redundant from complementary observations before global aggregation.

*(iii) Answer correctness $R_{\text{ans}}$.* This term evaluates task completion and is defined per task type. For counting and location reasoning (textual outputs),

$$R_{\text{ans}}^{\text{QA}} = \mathbb{I}[\hat{y} = y]. \tag{12}$$

For grasping (coordinate output $\hat{y} = (\hat{u}, \hat{v})$ with ground truth $y = (u, v)$), we use a distance-shaped reward

$$R_{\text{ans}}^{\text{grasp}} = \max\Big(0,\, 1 - \frac{\|\hat{y} - y\|_2}{d_{\max}}\Big), \tag{13}$$

where $d_{\max}$ is a normalization radius. While $R_{\text{ground}}$ and $R_{\text{overlap}}$ shape intermediate spatial reasoning, $R_{\text{ans}}$ enforces correctness of the final output.

## 5 EXPERIMENTS

### 5.1 EXPERIMENTAL SETUP

**Baselines and Protocol.** We compare **CoRL** with (i) proprietary VLMs: GPT-5 OpenAI (2025), Gemini-2.5-Pro Comanici et al. (2025), and Doubao-Seed-1.6 Seed (2025); (ii) open-source

Table 1: **Performance on the Ego2World-Bench (E2W-Bench).** This table groups tasks into Reasoning and Perception categories, with top two performances highlighted and rows alternately colored. **Highlighting**: **Top performance** (bold) and second best performance (underlined).

| Model | Reasoning (Acc.) | | | | Perception (Score) | | |
|---|---|---|---|---|---|---|---|
| | E2W-1 | E2W-2(S) | E2W-2(R) | Avg. | E2W-3(S) | E2W-3(R) | Avg. |
| *Closed Source Models* | | | | | | | |
| GPT-5 | 42.5 | 48.5 | 72.5 | 54.50 | 50.43 | 12.02 | 31.23 |
| Doubao-Seed-1.6 | 35.0 | 40.0 | 46.0 | 40.33 | 16.60 | 5.26 | 10.93 |
| Gemini-2.5-Pro | 32.5 | 42.5 | 50.0 | 41.67 | 35.98 | 10.15 | 23.07 |
| *Open Source Models* | | | | | | | |
| GLM-4.5v | 34.5 | 29.0 | 56.0 | 39.83 | 2.78 | 0.84 | 1.81 |
| SpaceQwen-3B | 21.5 | 6.0 | 60.0 | 29.17 | 16.15 | 5.07 | 10.61 |
| LLaMA-3.2-11b-vision-instruct | 16.5 | 12.5 | 17.5 | 15.50 | 7.78 | 3.38 | 5.58 |
| Qwen2.5VL-3B | 22.0 | 15.5 | 58.0 | 31.83 | 24.08 | 7.48 | 15.78 |
| Qwen2.5VL-7B | 17.0 | 17.0 | 64.5 | 32.83 | 28.83 | 5.78 | 17.31 |
| Qwen2.5VL-32B | 21.5 | 28.0 | 37.0 | 28.83 | 31.25 | 9.16 | 20.21 |
| *CoRL Variants* | | | | | | | |
| SFT-3B | 47.0 | 63.0 | 84.0 | 64.67 | 93.00 | 42.06 | 67.53 |
| RL-ZERO-3B | 23.0 | 39.5 | 83.5 | 48.67 | 50.63 | 8.02 | 29.33 |
| CoRL-3B | 59.0 | 75.5 | 86.0 | 73.50 | **96.30** | 41.82 | 69.06 |
| SFT-7B | 44.5 | 88.0 | 84.5 | 72.33 | 90.99 | 40.76 | 65.88 |
| RL-ZERO-7B | 16.0 | 56.0 | 82.5 | 51.50 | 92.60 | 11.65 | 52.13 |
| CoRL-7B | **61.0** | **97.0** | **90.0** | **82.67** | 95.69 | **44.32** | **70.01** |

VLMs: GLM-4.5V Team et al. (2025), Qwen2.5-VL-32B Bai et al. (2025), LLaMA-3.2-11b-vision-instruct Dubey et al. (2024), SpaceQwen-3B Yang et al. (2025); and (iii) our model variants on Qwen2.5-VL-Instruct Bai et al. (2025) backbones (3B/7B): SFT-only, RL-from-scratch (RL-ZERO). All CoRL variants are trained on the E2W training set. All models are evaluated under the same prompts and input aggregation protocol. Complete training details, including hyperparameter settings and code, are available in our repository.

**Evaluation Benchmarks and Metrics.** We conduct our primary evaluation on the proposed **E2W-Bench**, which comprises the **Counting**, **Location Reasoning**, and **Grasping** tasks detailed in Section 3. For the QA-based tasks (Counting and Location Reasoning), we report exact match accuracy. For **Grasping**, we report a normalized score from 0 to 100, calculated based on the Euclidean distance between the predicted and ground-truth coordinates, consistent with the task's reward function defined in our methodology. To further assess the generalization capabilities of our approach, we also report performance on the external Where2Place dataset, a standard testbed for spatial reasoning. A detailed description of dataset statistics, implementation specifics, and evaluation protocols is provided in the Appendix C.

## 5.2 MAIN RESULTS ON E2W-BENCH

Our main findings on the E2W-Bench are summarized in Table 1. The results show a clear performance hierarchy across the different model categories. In a zero-shot setting, proprietary models like GPT-5 establish the strongest baseline, significantly outperforming open-source VLMs, which generally struggle with the benchmark's complex multi-view demands. Our CoRL variants, which are fine-tuned on E2W-Bench training set, demonstrate the efficacy of our proposed training pipeline.

Table 2: **Ablation of CVSR Components**. Results of CoRL-7B on E2W-Bench.

| Setting | E2W-1 | E2W-2-Sim | E2W-3-Sim |
|---|---|---|---|
| CoRL (CVSR) | **61.0** | **97.0** | **95.69** |
| – w/o $R_{\text{ans}}$ | 10.5 | 15.5 | 40.32 |
| – w/o $R_{\text{ground}}$ | 50.5 | 90.5 | 74.32 |
| – w/o $R_{\text{overlap}}$ | 56.5 | 90.0 | 84.31 |
| – w/o $R_{\text{format}}$ | 58.5 | 93.0 | 89.31 |
| – SFT-only | 44.5 | 88.0 | 90.99 |

Table 3: **Single-view vs. Multi-view Inputs**. Performance on E2W-Bench.

| Setting | E2W-1 | E2W-2-Sim |
|---|---|---|
| *7B Backbone* | | |
| Single-view | 34.0 | 54.0 |
| Multi-view | **61.0** | **97.0** |
| *3B Backbone* | | |
| Single-view | 36.5 | 51.5 |
| Multi-view | **59.0** | **75.5** |

The SFT-only models establish a very strong performance level, while the full CoRL (SFT+RL) framework consistently achieves the highest scores across both 3B and 7B model scales. Notably, the *RL-from-scratch* ablation performs poorly, underscoring the necessity of a supervised warm-up phase to stabilize optimization and provide a reliable initialization. This observation further highlights the complementary nature of SFT and RL, where supervised fine-tuning imparts essential reasoning priors and reinforcement learning subsequently refines them toward task-specific objectives.

A deeper analysis across task categories reveals distinct performance patterns. On the **Reasoning** tasks (E2W-1 and E2W-2), which require aggregating abstract information across views, the largest models show some inherent capability. However, the RL stage of our CoRL framework provides a crucial advantage, refining the model's ability to resolve cross-view ambiguities and synthesize a coherent global state, as evidenced by the superior scores of CoRL models. The most pronounced gap emerges in the **Perception Grounding** task (E2W-3). Here, nearly all zero-shot models fail to produce reliable coordinates, exposing a weakness in fine-grained spatial grounding. In contrast, our SFT and CoRL models excel—thanks to explicit supervision in our dataset and, for CoRL, the targeted policy optimization from CVSR's grounding and consistency rewards, which shape the model toward physically precise and reliable outputs.

## 5.3 Ablation Studies

To validate our design choices, we conduct a series of ablation studies. We first analyze the contribution of each component within our CVSR design, and then investigate the performance gap between reasoning from distributed ego-centric views versus a single, privileged global view. These studies provide a deeper understanding of how individual reward signals and viewpoint configurations jointly influence the emergence of robust cross-view reasoning.

**Impact of CVSR Components.**  We ablate each component of the Cross-View Spatial Reward (CVSR) to assess its individual contribution (Table 2). Removing the answer correctness reward ($R_{\text{ans}}$) causes a catastrophic drop in E2W-1 accuracy from **61.0%** to 10.5%, showing that intermediate shaping signals alone cannot ensure correct final solutions. Eliminating the grounding reward ($R_{\text{ground}}$) reduces E2W-3-Sim by over 21 points, confirming its role in aligning symbolic reasoning with precise object locations. Dropping the overlap reward ($R_{\text{overlap}}$) weakens counting and relational reasoning by impairing cross-view object consistency. Finally, the format reward ($R_{\text{format}}$) provides smaller but meaningful gains by enforcing structured outputs and stabilizing optimization. Together these results show that only the full CVSR—balancing correctness, grounding, consistency, and structural integrity—can robustly guide multi-view reasoning and embodied performance.

**Multi-View Fusion vs. Global View Perception.**  We next investigate a critical trade-off: is it better to process multiple, high-resolution ego-centric views or a single, downsampled global view under a fixed image token budget? Table 3 shows that the multi-view configuration consistently outperforms the global view. For instance, on E2W-2-Sim, our multi-view approach achieves **97.0%** accuracy, significantly surpassing the global view's **85.5%**. This result suggests that for complex spatial reasoning, preserving high-resolution local details is more critical than having a complete but coarse global context. While downsampling makes the global view computationally tractable, it degrades crucial information about small objects and fine-grained spatial relations. Our CoRL

Table 4: Where2Place Results

| Model | Score ↑ |
|---|---|
| SpaceLLaVA | 11.8 |
| RoboPoint | 46.8 |
| Molmo-7B | 45.0 |
| **CoRL-7B (Ours)** | **50.9** |

Table 5: Real-World Robot Evaluation: Success Rates (%)

| Tasks with collaborative spatial reasoning | RoboPoint | **Ours** |
|---|---|---|
| Grasp the blue block that is farthest from the strawberry and place it on the red block | 0.00 | **65.00** |
| Pick up the carambola that is aligned with the banana and place it on the red block | 0.00 | **30.00** |

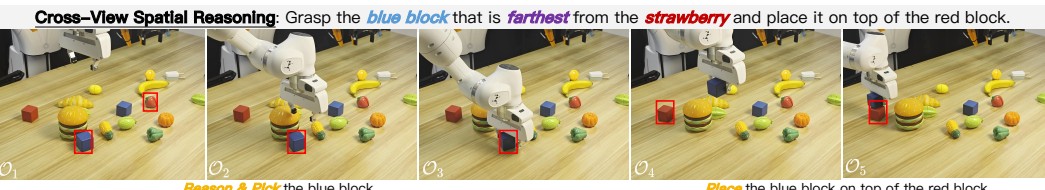

Figure 4: Illustrative demonstrations of CoRL-7B in real-world evaluation.

framework, in contrast, demonstrates its ability to effectively fuse detailed partial views to achieve superior performance, highlighting the primacy of information density over contextual completeness within a fixed computational budget.

### 5.4 EXTERNAL BENCHMARK: WHERE2PLACE.

We further test single-image spatial grounding on **Where2Place** (Yuan et al., 2024), which evaluates fine-grained point prediction from a single RGB view (Table 4). CoRL-7B attains 50.9 points, outperforming RoboPoint (46.8) and Molmo-7B (45.0). This shows that cross-view training with CVSR not only benefits multi-view reasoning but also strengthens single-image pointing ability.

### 5.5 REAL-WORLD EVALUATION

We validate the effectiveness of CoRL-7B through real-world experiments, where two Franka Research 3 arms are mounted on a Realman mobile base, each equipped with an Intel RealSense D435 RGB-D camera with calibrated extrinsics. The multi-view observations are fused to support spatial localization and grasping, while CoRL-7B predicts target positions for manipulation.

As shown in Table 5, CoRL achieves strong performance across different tasks, with success rates of 65% on the blue block picking task and 30% on the carambola alignment task, significantly outperforming the RoboPoint baseline (0% on both tasks). These results highlight the robustness of our method in handling visual ambiguity, such as occlusions and distractors with similar colors.

Qualitative rollouts in Figure 4 further illustrate CoRL's reasoning capability. The robot successfully infers the correct target object across multiple views, picks it up despite clutter and distractors, and places it accurately at the instructed location. Together, the quantitative and qualitative evaluations confirm that CoRL is effective for real-world manipulation, demonstrating reliable reasoning ability.

## 6 CONCLUSION

In this work, we addressed the challenge of collaborative spatial reasoning across distributed ego-centric views. We proposed **CoRL**, a two-stage SFT→RL framework that fine-tunes Vision–Language Models (VLMs) with a novel Cross-View Spatial Reward (CVSR) to encourage multi-view fusion and precise spatial grounding. Extensive experiments on the newly introduced **E2W-Bench** show that CoRL consistently surpasses strong proprietary and open-source baselines. This performance gain derives from the synergy of SFT, which establishes a strong initial policy, and CVSR-guided RL, which reinforces cross-view consistency and grounding. By coupling a challenging benchmark with a principled training methodology, CoRL charts a path toward multi-agent embodied systems capable of constructing coherent world models from partial and distributed perception.

ETHICS STATEMENT

This work makes use of the publicly available Ego4D Grauman et al. (2022)dataset and the ManiSkill3 simulation platform Tao et al. (2024). The dataset was collected and released in accordance with ethical research practices, and all experiments conducted in simulation do not involve human or animal subjects. Our study focuses on improving the capabilities of vision-language models, and we do not foresee any direct ethical concerns, including issues of privacy, safety, or fairness. We believe the potential benefits of advancing embodied AI research outweigh the minimal risks, and no conflicts of interest or ethical violations are associated with this work.

REPRODICIBILITY STATEMENT

We have taken extensive measures to ensure the reproducibility of our work. All code related to data preprocessing, model training, and evaluation will be released, and the trained models will be hosted on Hugging Face for public access. During the review process, we will provide an anonymous link to the complete implementation and resources. The processed data used in our experiments will also be made available to facilitate replication. Furthermore, detailed experimental settings, including hyperparameters, training schedules, and environment configurations, are documented in the appendix and supplementary materials. Together, these efforts are intended to enable independent researchers to fully reproduce and verify our results.

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

# Supplementary Material

## A   USE OF LARGE LANGUAGE MODELS (LLMs)

We used large language model (GPT-5) as an assistive tool in two ways: (1) for writing assistance, including language editing and improving the clarity of the manuscript, and (2) for technical support during code environment setup and debugging, particularly when resolving environment-related errors. The model was not used for generating research ideas, designing methodologies, conducting experiments, or analyzing results. All outputs from the LLM were manually verified by the authors, and final decisions regarding both the research content and the manuscript were made by the authors. The authors take full responsibility for the entirety of this work.

## B   PROBLEM SETUP & NOTATION

In this section, we formally define the problem setup and introduce the notation used throughout the paper. We focus on the collaborative spatial reasoning task within multi-agent embodied systems, where multiple agents contribute to the construction of a shared understanding of a scene from partial observations.

### B.1   PROBLEM SETUP

We consider a system of $N$ agents, each represented by an individual agent $a_i$ for $i \in \{1, \ldots, N\}$. Each agent $a_i$ perceives the environment through a limited, ego-centric viewpoint, denoted by $\mathcal{V}_i$. The environment $\mathcal{E}$ is a spatial scene consisting of various objects, denoted by $\mathcal{O} = \{o_1, o_2, \ldots, o_M\}$, where $M$ is the total number of objects in the scene. Each object $o_j$ has associated properties, such as position, orientation, and object class, which we denote by $\mathbf{p}_j = (x_j, y_j, z_j)$, $\mathbf{r}_j = (\theta_j, \phi_j)$, and $c_j$ respectively.

The goal of the system is to enable the agents to collaboratively reason about the scene, including the localization of objects, the spatial relationships between objects, and the execution of spatially grounded actions. To achieve this, each agent $a_i$ shares its ego-centric observation $\mathcal{V}_i$ with a central Vision–Language Model (VLM), which integrates the individual observations to build a global understanding of the scene.

The task is formalized as a spatial reasoning problem, where the agents must complete a set of spatial tasks, such as object localization, relationship inference, and action planning, by collectively reasoning over their partial observations. Specifically, we aim to develop a system where the agents' observations are integrated effectively to:

- Localize objects across different viewpoints.
- Infer spatial relationships between objects.
- Plan and execute actions based on a shared understanding of the scene.

### B.2   NOTATION

We define the following notation used in this paper:

- $\mathcal{A}$: Set of agents, where $\mathcal{A} = \{a_1, a_2, \ldots, a_N\}$, and each agent $a_i$ has its own ego-centric observation $\mathcal{V}_i$.
- $\mathcal{O}$: Set of objects in the scene, $\mathcal{O} = \{o_1, o_2, \ldots, o_M\}$.
- $\mathcal{V}_i$: Ego-centric observation of agent $a_i$, containing visual and linguistic information relevant to the agent's current view of the scene.
- $\mathbf{p}_j$: Position of object $o_j$, represented as a 3D vector $(x_j, y_j, z_j)$.
- $\mathbf{r}_j$: Orientation of object $o_j$, represented by angles $(\theta_j, \phi_j)$.
- $c_j$: Class of object $o_j$, which can be one of a predefined set of object categories.

- $y_i$: The set of actions available to agent $a_i$, such as move, grasp, or navigate.

- $\mathcal{R}_i$: Reward function for agent $a_i$, which quantifies the success of the agent's actions with respect to the spatial reasoning task.

- $Q$: A central Vision–Language Model (VLM) that integrates all ego-centric observations to form a global understanding of the scene.

- $s$: The global scene state, which is a combination of the individual observations from all agents.

### B.3 TASK DEFINITION

We define the collaborative spatial reasoning task as a sequential decision-making process where each agent must select an action from the set $\mathcal{A}_i$ based on its perception $\mathcal{V}_i$ and the shared scene information. The goal is to maximize the collective performance across all agents, measured by a global reward function that takes into account the accuracy of object localization, the correct inference of spatial relationships, and the successful execution of grounded actions.

The multi-agent system operates under the assumption of partial observability, meaning that each agent only has access to a limited subset of the scene and must rely on the collaboration of other agents to complete the task. The system must overcome the challenges of occlusions, incomplete observations, and ambiguous spatial relationships to successfully execute actions in a dynamic environment.

### B.4 ASSUMPTIONS

The following assumptions are made for the proposed problem setup:

- Each agent has access to an ego-centric camera or sensor that provides partial observations of the scene.

- Agents can communicate with each other to share observations and jointly reason about the spatial layout of the environment.

- The central VLM has access to all agent observations and coordinates the integration of these inputs for collaborative reasoning.

- The agents' actions are executed in discrete time steps, and the environment responds to each action with a new state.

## C E2W BENCHMARK DETAILS

In this section, we provide detailed information about the E2W benchmark, which consists of both simulated and real-world data. The dataset includes over 100k simulated samples and more than 6k real-world samples. The benchmark is organized into three tasks, each with a distinct focus on collaborative spatial reasoning. Below, we describe the structure and data distribution for each task.

### C.1 DATASET OVERVIEW

The E2W benchmark consists of the following components:

- **Simulated Data**: A total of over 100k samples were collected from simulations, which provide a diverse set of scenes and spatial configurations for training and evaluation.

- **Real-World Data**: Over 6k samples were gathered from real-world environments, capturing the complexity of physical spaces and sensor noise that is typical in practical scenarios.

### C.2 TASK BREAKDOWN

The E2W benchmark includes three distinct tasks designed to evaluate different aspects of collaborative spatial reasoning:

- **Task 1 (Counting Task)**: This task focuses on the ability of the agents to count the number of objects within a scene. It is solely based on the simulated data, as real-world data for this task was not available.

- **Task 2 (Object Localization)**: This task evaluates the agents' ability to localize objects within the scene. Both simulated and real-world data are used, with the real-world data providing additional complexity due to sensor noise and occlusions.

- **Task 3 (Spatial Relationship Inference)**: In this task, agents are required to infer spatial relationships between objects (e.g., proximity, occlusion). Like Task 2, this task uses both simulated and real-world data.

## C.3 DATASET SPLITS

For each task, we carefully curated the dataset as follows:

- **Test Set**: For each task, we selected 200 high-quality samples to form the test set, ensuring that it covers a wide range of challenging scenarios.

- **Cold-Start Set (COT)**: We prepared a cold-start set consisting of 1000 samples, which is used to initialize the model before training. This set includes diverse configurations that enable the agents to begin learning without prior knowledge of the environment.

- **Training Set**: The remaining samples were used for training, with 90k samples from the simulated data contributing to the training set. The real-world data (6k samples) was integrated into the training set, but it is more sparsely used compared to the simulated data, ensuring a balance between generalizability and real-world applicability.

## C.4 SUMMARY OF DATASET DISTRIBUTION

| Task | Simulated Data | Real-World Data | Total Data |
|------|----------------|-----------------|------------|
| Task 1 (Counting) | 30k | 0 | 30k |
| Task 2 (Localization) | 35k | 40k | 75k |
| Task 3 (Inference) | 35k | 40k | 75k |

Table 6: E2W Benchmark Dataset Distribution for Each Task

## C.5 DATA QUALITY AND SELECTION CRITERIA

Each task in the benchmark was carefully designed to cover a range of real-world challenges in collaborative spatial reasoning. To ensure high data quality, we followed strict selection criteria for both simulated and real-world data:

- The simulated data was generated using a variety of scene configurations, object types, and spatial relationships to create a comprehensive and diverse training set.

- For the real-world data, we selected scenarios with clear object localization, minimal occlusions, and representative spatial relationships to maximize the relevance of the data for evaluating real-world performance.

- We prioritized edge cases and challenging scenarios for the test sets, ensuring that they push the limits of the agents' reasoning abilities and provide meaningful benchmarks for model performance.

## C.6 DATA COLLECTION

**Simulation Data**    All task data are collected in a high-fidelity simulation environment built upon RoboFactory Qin et al. (2025) and ManiSkill3 Tao et al. (2024), which provide diverse scenes like RoboCasa Nasiriany et al. (2024), articulated objects, and multi-agent configurations. We curate over 15,000 multi-agent samples across dozens of spatial layouts, each scene populated with a rich combination of manipulable objects under varied configurations. For every instance,

synchronized global and egocentric camera views are recorded to support collaborative reasoning. Ground-truth annotations, including object counts, spatial relations, and manipulation-relevant attributes, are automatically derived from simulator metadata and physics engines, followed by human verification to ensure semantic consistency. This pipeline ensures that the resulting dataset captures the complementary demands of multi-view spatial reasoning, ranging from global aggregation to relational understanding and action grounding.

**Real-World Data**    To complement simulation environments with naturalistic observations, we leverage the Ego4D dataset Grauman et al. (2022), a massive-scale egocentric video corpus spanning 74 worldwide locations across 9 countries, with over 3,670 hours of daily-life recordings. Ego4D offers unconstrained visual contexts that reflect the challenges of embodied perception in realistic human environments. For data preparation, we first extract video frames at uniform intervals to obtain temporally diverse samples. We then apply strong vision backbones, including Region Attention Masking (RAM) Zhang et al. (2023b) for object-level proposals and DINO Caron et al. (2021) for robust feature alignment, to automatically annotate object instances and spatial relations. Through this pipeline, we curate two subsets aligned with our benchmarks: **E2W-2-Real** (relational reasoning) and **E2W-3-Real** (grasping-oriented perception), each containing approximately 30k samples, yielding a total of 60k real-world instances. This large-scale collection bridges the gap between simulation and reality by introducing natural visual noise, diverse object appearances, and unconstrained scene dynamics.

# D    METHOD DETAILS (CoRL)

In this section, we provide a detailed description of the proposed Collaborative Spatial Reasoning Reinforcement Learning (CoRL) framework. We outline the training objective, pipeline, and key components of the approach, which enable effective multi-agent collaborative spatial reasoning.

## D.1    TRAINING OBJECTIVE & PIPELINE

The training objective for our proposed CoRL framework is designed to guide the agents towards improving their collaborative spatial reasoning capabilities. The main goal is to maximize the agents' ability to effectively integrate their partial, ego-centric observations and reason about the global spatial scene. To achieve this, we combine reinforcement learning (RL) with a novel Cross-View Spatial Reward (CVSR) function, which shapes the learning process by encouraging the agents to perform tasks that require accurate spatial localization, object relationships inference, and collaborative decision-making.

### D.1.1    REINFORCEMENT LEARNING SETUP

We adopt a reinforcement learning setup, where each agent interacts with its environment and receives feedback in the form of a reward signal. The environment consists of a spatial scene with multiple agents, and each agent's objective is to complete a set of spatial reasoning tasks, such as object localization and relationship inference.

Each agent $a_i$ receives a partial observation $\mathcal{V}_i$ of the scene, which includes both visual and linguistic information. Based on this observation, the agent selects an action $a_i(t)$ from a predefined set of actions $y_i$. The agent's action affects the state of the environment, and the environment responds by providing a new state and a corresponding reward $r_i(t)$, which is calculated by the Cross-View Spatial Reward (CVSR) function.

The overall training objective is to maximize the expected cumulative reward for each agent:

$$J(\theta) = \mathbb{E}\left[\sum_{t=0}^{T} \gamma^t r_i(t)\right]$$

where $T$ is the total number of time steps in the task, $\gamma$ is the discount factor, and $r_i(t)$ is the reward signal at time step $t$.

## D.2 Hyperparameters for Reward and CVSR

In our CoRL framework, we define several hyperparameters for the reward function and Cross-View Spatial Reward (CVSR) that guide the agents' training process. These weights are carefully tuned to balance the different components of the task, ensuring that the agents effectively learn to perform spatial reasoning across multiple agents and viewpoints.

### D.2.1 CVSR Components Weights

The Cross-View Spatial Reward (CVSR) function is designed to guide agents in overcoming challenges like occlusions, incomplete observations, and the need for integration across multiple viewpoints. The CVSR reward is composed of three components:

$$r_i(t) = w_{ans} \cdot \mathcal{L}_{ans} + w_{loc} \cdot \mathcal{L}_{loc} + w_{fusion} \cdot \mathcal{L}_{fusion}$$

where the components are:

- $w_{ans} = 0.7$: The weight for Answering (the ability to correctly complete the task based on spatial reasoning). This is the most important component, as it directly measures the success of the agent in solving the task.
- $w_{loc} = 0.1$: The weight for Localization (the ability to accurately localize objects in the scene). While important, this is a lower priority in comparison to answering tasks, as it is a fundamental skill that supports other reasoning tasks.
- $w_{fusion} = 0.2$: The weight for Fusion of Observations (the integration of ego-centric views). This component ensures the agents combine their partial observations to build a more complete understanding of the environment, and is essential for collaborative spatial reasoning.

These weights are chosen to emphasize the importance of answering correctly, while still ensuring that localization and fusion of observations are effectively learned.

### D.2.2 Training Pipeline

The training pipeline for the CoRL framework follows a two-phase process: Supervised Fine-Tuning (SFT) followed by Reinforcement Learning (RL). The pipeline is as follows:

- **Phase 1: Supervised Fine-Tuning (SFT)**: In this phase, the agents are first pre-trained using supervised learning. The goal is to initialize the model with basic spatial reasoning skills, using labeled data from the training set. The agents learn to perform tasks such as object localization and relationship inference based on ground truth labels.
- **Phase 2: Reinforcement Learning (RL)**: After the initial fine-tuning, the agents enter the RL phase, where they learn to improve their performance through interactions with the environment. The CVSR function is used to guide the agents' actions and refine their spatial reasoning capabilities. During this phase, agents iteratively adjust their policies to maximize the cumulative reward.

The overall training process is summarized in Algorithm 1.

# E  Real-World Robotic Evaluation

We implement our real-world evaluation on manipulation platform equipped with two Franka Research 3 arms and a Realman mobile base. Each arm is paired with an Intel RealSense D435 RGB-D camera beside the robotic arm, and all cameras are extrinsically calibrated to a common base frame.

In the open-loop execution mode, CoRL-7B processes synchronized multi-view RGB images and predicts the 2D target location for the instructed action. The 2D coordinates are first fed into SAM2 Ravi et al. (2024) to generate a segmentation mask, which filters the target object's point cloud from the RGB-D stream of the D435. The extracted point cloud is then passed to AnyGrasp Fang et al.

---

**Algorithm 1** CoRL Training Pipeline

---

1: **Input:** Initial model parameters $\theta$, training dataset $\mathcal{D}$, CVSR weights $w_1, w_2, w_3$
2: **Phase 1: Supervised Fine-Tuning**
3: **for** each agent $a_i$ in $\mathcal{A}$ **do**
4:     Initialize agent policy $\pi_i$ using labeled data from $\mathcal{D}$
5:     Fine-tune agent using supervised learning
6: **end for**
7: **Phase 2: Reinforcement Learning**
8: **for** each agent $a_i$ in $\mathcal{A}$ **do**
9:     Initialize RL agent with fine-tuned policy $\pi_i$
10:     **for** each episode in $\mathcal{E}$ **do**
11:         Collect observations $\mathcal{V}_i$ from environment
12:         Select action $a_i(t)$ using $\pi_i$
13:         Execute action and receive reward $r_i(t)$ from CVSR
14:         Update policy $\pi_i$ using RL algorithm
15:     **end for**
16: **end for**

---

(2023), which predicts a feasible grasp pose in the camera coordinate frame. Using the pre-calibrated camera extrinsics, the grasp pose is transformed into the coordinate system of the selected robotic arm. Then, we use the Deoxys library Zhu et al. (2022) to interact with the Franka Control Interface.

For placement, CoRL-7B outputs a 2D placement location, which is converted into 3D coordinates using the depth data. This 3D point is similarly transformed into the robot's coordinate frame, and the arm follows an open-loop trajectory to release the object at the designated position.

This setup allows directly connect vision–language inference with physical manipulation through SAM2-based segmentation and AnyGrasp-based grasp synthesis, without closed-loop corrections. It highlights the model's ability to produce actionable spatial predictions from multi-view observations.

## F   Limitations & Future Work

We acknowledge several limitations that offer promising avenues for future research. First, the current CoRL framework relies on a centralized VLM that processes all ego-centric views simultaneously. While effective for a small number of agents, its scalability to scenarios involving a large fleet of agents warrants further investigation into more efficient, perhaps hierarchical or message-passing-based, fusion strategies. Second, our experiments primarily focus on quasi-static scenes. Extending the framework to highly dynamic environments where agents and objects are in concurrent motion would require incorporating temporal reasoning and is a significant future challenge.

This work opens several exciting directions. Future research could explore decentralized architectures where agents learn to communicate condensed, relevant information rather than sharing raw visual data. Another compelling avenue is the extension of CoRL to long-horizon, multi-step collaborative tasks, moving beyond single-step perception and grasping towards complex manipulation and navigation. Ultimately, we believe our findings provide a robust foundation and a principled methodology for developing the next generation of collaborative embodied AI.

### F.1   Discussion

**Implications.**   Our comprehensive experiments convey a clear, overarching message: while model scale and pretraining provide a foundation, they are insufficient for mastering the complexities of collaborative embodied reasoning. The success of our SFT→RL pipeline highlights a crucial insight: robust spatial intelligence emerges from a structured curriculum that first bootstraps foundational, in-domain knowledge (via SFT) and then refines nuanced, collaborative behaviors through targeted reward shaping (via our CVSR). The reinforcement learning stage proves essential for teaching the model not just **what** the right answer is, but **how** to systematically derive it by grounding its reasoning in visual evidence and maintaining cross-view consistency. Furthermore, our ablation on input modality reveals a non-obvious trade-off between information density and contextual breadth,

suggesting that under fixed computational budgets, multiple high-resolution local views can be more valuable than a single, coarse global one.

## F.2 ADDITIONAL IMPLEMENTATION DETAILS

For detailed code and further information, please visit our repository: CoRL Repository

