# OpenReview forum: "Ego to World: Collaborative Spatial Reasoning in Embodied Systems via Reinforcement Learning"
_ICLR.cc/2026/Conference — ICLR 2026 Conference Withdrawn Submission_

### Official Review · Reviewer_ecx3 · 2025-10-15

**Soundness:** 3
**Presentation:** 2
**Contribution:** 2
**Rating:** 2
**Confidence:** 5

**Summary:**

The paper presents Ego-to-World (E2W), a benchmark for evaluating vision–language models on multi-view spatial reasoning, including counting, relational localization, and view-specific grasping. It also introduces CoRL, which integrates chain-of-thought fine-tuning with reinforcement learning to deliver consistently superior results over competitive baselines.

**Strengths:**

- This paper introduces the E2W benchmark to rigorously evaluate multi-view spatial reasoning in VLMs across global counting, relational localization, and action-oriented grasping with view-specific coordinates.
- The CoRL framework and reward design are intuitive. And the paper demonstrates consistent gains over strong proprietary and open-source baselines on reasoning and perception-grounding metrics; ablations substantiate the necessity of each CVSR component.

**Weaknesses:**

-  *Problem motivation and setting rationale:* The paper does not convincingly justify when multi-view, per-agent ego-centric fusion is preferable to constructing a single global view, especially in real-world multi-robot systems where calibrated multi-camera rigs are available. While the authors show that global-view inputs under token downsampling perform worse, this seems like a resource-allocation artifact rather than a fundamental limitation. Please clarify the operational regimes where the proposed setting is necessary (e.g., bandwidth/latency constraints, or decentralization requirements).
- *Novelty relative to recent cross-view alignment:* Spatial reasoning per se is important but not novel, and several recent works address cross-view/ego-view alignment with sequence-level reasoning. For instance, “Steering Visuomotor Policy in Open Worlds via Cross-View Goal Alignment” (arXiv:2503.02505) even discusses long-horizon cross-view/ego-view alignment and introduces auxiliary objectives that appear conceptually similar to the proposed Cross-View Spatial Reward (CVSR). The paper should position CVSR more explicitly against such alignment losses, detailing what is new (e.g., reward shaping granularity, step-to-evidence linkage, multi-task applicability) and why it is necessary beyond existing auxiliary alignment strategies.

**Questions:**

- I am curious about the specific role of chain-of-thought (CoT) in the reported gains. Could similar improvements be achieved with straightforward fine-tuning under your setting? In traditional decision-making, for example, “Scalable Multi-Task Reinforcement Learning for Generalizable Spatial Intelligence in Visuomotor Agents,” task success alone can enhance spatial reasoning performance—how does CoT compare against such baselines in your experiments?

---

### Official Review · Reviewer_nNNd · 2025-10-25

**Soundness:** 3
**Presentation:** 3
**Contribution:** 3
**Rating:** 6
**Confidence:** 4

**Summary:**

The paper tackles collaborative spatial reasoning in multi-agent embodied systems by introducing the Ego-to-World (E2W) benchmark and CoRL training framework. The novel Cross-View Spatial Reward (CVSR) aligns reasoning steps with visual grounding, overlap-aware instance resolution, and final-answer correctness. Experiments show CoRL consistently outperforms strong proprietary and open-source VLMs on E2W across reasoning and perception-grounding metrics, with ablations confirming each CVSR component’s contribution. The approach generalizes to Where2Place and demonstrates real-world multi-robot manipulation with calibrated multi-camera rigs, achieving reliable cross-view localization and grasp-and-place execution.

**Strengths:**

- Originality: Introduces a key robotics problem—multi-agent, multi-view embodied spatial reasoning—and presents the novel E2W benchmark and the Cross-View Spatial Reward (CVSR).
- Quality: The E2W benchmark is well defined, practical, and accurately captures multi-view spatial reasoning. The CVSR is straightforward and efficient.
- Clarity: The problem formulation, method overview, training objective, and reward design are clearly specified, with equations and figures, within a transparent pipeline (SFT initialization, GRPO updates, CVSR composition). Dataset construction (simulation and real), task definitions, metrics, and the evaluation protocol are described reproducibly.
- Significance: The work establishes a practical path toward world-centric scene understanding from distributed views, addressing a key bottleneck in collaborative embodied AI. By linking reasoning traces to visual grounding and actionable 2D coordinates, it advances cross-view fusion for manipulation and offers a benchmark and methodology likely to catalyze further research in multi-agent spatial reasoning.

**Weaknesses:**

- The real-world subset includes only single-image, single-agent samples; multi-view and multi-agent settings are available only in simulation.
- The real-world evaluation setup is overly simple, with a black cloth background and uniform lighting. VLMs should also be tested in more in-the-wild, cluttered scenes with varied lighting and occlusion to assess robustness.

**Questions:**

- Regarding the first weakness, does the data-collection pipeline support annotation in *real-world*, multi-view settings? How are object correspondences across views reliably established—for example, verifying that the two blue boxes in View 1 and View 2 refer to the same object?
- Does the VLM require camera intrinsics/extrinsics as input, or can it rely on equivalent information (e.g., robot proprioception for wrist-mounted cameras)?

---

### Official Review · Reviewer_tmqR · 2025-10-29

**Soundness:** 2
**Presentation:** 1
**Contribution:** 2
**Rating:** 2
**Confidence:** 3

**Summary:**

This paper introduces  Ego-to-World (E2W) benchmark and a corresponding method CoRL for finetuning VLMs for multi-view spatial reasoning. It is demonstrated by the experiments that the proposed CoRL is able to outperform open-source VLMs and proprietary models on the proposed benchmark. It is also demonstrated by CoRL is able to transfer to Where2Place, a single-view spatial grounding task, after trained on the E2W dataset.

**Strengths:**

1. The paper is easy to follow.

2. The proposed reward used for GRPO is able to guide the finetuning of a VLM to become better in terms of spatial reasoning, specifically counting, location reasoning and affordance prediction.

**Weaknesses:**

1. The proposed method uses a single VLM to handle multiple views, which disobey the claim that the paper address "collaborative" spatial reasoning.

2. With the above mentioned point, the paper is essentially proposing a multi-image spatial reasoning task and method, where the difficulties of the proposed tasks seem to be easier than already existing benchmark like MindCube [1]

3. The proposed method is simply using existing framework, i.e. SFT+GRPO, which has already been proved to be useful for several reasoning tasks that requires finetuning.


[1] Yin, B., Wang, Q., Zhang, P., Zhang, J., Wang, K., Wang, Z., Zhang, J., Chandrasegaran, K., Liu, H., Krishna, R. and Xie, S., 2025, June. Spatial mental modeling from limited views. In Structural Priors for Vision Workshop at ICCV'25.

**Questions:**

1. Has the author tried to transfer the learned model to benchmarks like MindCube [1] to see whether the proposed reward signal would still be useful?

2. Regarding the comparison between multi-view and global-view, why does the global-view image have to be low-resolution? If the claim was to limit the image token used, then it's not a fair comparison since multiple images would use more tokens intuitively.

3. In Figure 2, the first orange box in the top right corner shows that the model thinks there are 2 apples, while the final answer is 3. Is this the original output by the model? Or is it a handcrafted example?

---

### Official Review · Reviewer_YJMT · 2025-10-31

**Soundness:** 3
**Presentation:** 3
**Contribution:** 2
**Rating:** 6
**Confidence:** 4

**Summary:**

This paper introduces Ego-to-World (E2W), a new benchmark for evaluating collaborative spatial reasoning in multi-agent embodied systems, and proposes CoRL, a two-stage framework combining supervised Chain-of-Thought (CoT) fine-tuning with reinforcement learning (RL). The key innovation lies in the Cross-View Spatial Reward (CVSR), a structured reward that aligns reasoning steps with visual evidence, enforces cross-view consistency, and improves grounding in multi-view settings. The paper reports strong empirical results on the proposed E2W-Bench and external Where2Place benchmark, as well as in real-world robotic experiments, showing superior reasoning and grasping performance over strong baselines.

**Strengths:**

1. The paper defines the task of collaborative spatial reasoning under distributed ego-centric observations, which is parctically crucial as many real-world multi-robot and multi-camera systems inherently operate under partial, viewpoint-specific observations, and yet proper methods for such scenarios are underexplored in embodied AI.
2. The proposed goal to transform fragmented ego-centric views into a globally coherent and semantically consistent scene representation is novel and elegant. Such goal not only pushes visual spatial reasoning from single-view to multi-view, but also push VLMs to improve their spatial intelligence to understand a real-world scene with joint visual reasoning.
3. The E2W-Bench covers multiple tasks including global counting, relational reasoning, and grasping that jointly assess multi-view spatial reasoning in both simulation and real-world scenarios.
4. The Cross-View Spatial Reward (CVSR) introduces dense, interpretable reward components (grounding, overlap, and answer correctness) that jointly enforce cross-view visual reasoning, providing a new thread to tackle challenges in collaborative perception and multi-view sptial reasoning.

**Weaknesses:**

1. Although E2W covers diverse tasks, the visual data shown in the paper is relatively clean and uncluttered. The presented static-tabletop setups of real-world setting and simulation scene in Figure 2.c contain sparse objects with few occlusions and background noise. This simplicity may underrepresent challenges encountered in real-world embodied settings, thus more complex and visually noisy scenes would make the benchmark more representative of the embodied AI scenarios that the paper aims to address.
2.
   The paper focuses heavily on quantitative results, but provides little qualitative analysis of *how* the model performs cross-view reasoning. The absence of case studies, visualization of reasoning traces, and examples of both successful and failed predictions limits interpretability. Presenting details like CoT reasoning paths would help enhance its presentation and clarity.
3. Despite diverse tasks, the evaluated settings remain *passive* and *single-step*. The model only needs to produce an answer or predict 2D coordinates, without active interaction, or multi-turn decision-making, which simplifies the inherent difficulty of real-world multi-agent Embodied AI reasoning tasks.
4.
   Some notation inconsistency reduce the clarity. For example, the CVSR components are described as Grounding Reward, Overlap Accuracy, and Answer Correctness in Section 4.4, but later are presented as Answering, Location, and Fusion of Observations in Section D.2.1.

**Questions:**

1. In Line 302, how do the authors define “*unique object instances*” when defining the Overlap Accuracy component of CVSR. Does it refer strictly to semantically distinct objects (e.g., individual apples on a table), or does it also include fine-grained visual details (e.g., texture differences or background marks)? Furthermore, how are these instances annotated, like through human labeling, or a automatic rule-based procedure?

2. How does CoRL *scale* as the number of agents/views increases? While more views may provide richer context, they also introduce combinatorial complexity in alignment and reasoning. Has the paper examined whether the scaling will affect the model performance?

3. In the E2W-2 tasks reported in Table 1, most models exhibit higher performance in real-world data compared to simulation. This trend is counterintuitive, as simulation environments usually provide cleaner and more controlled inputs. Could the authors provide more reasons for this discrepancy.

4. Although CVSR is designed to promote cross-view reasoning through components $R_{\text{ground}}$, $R_{\text{overlap}}$, and $R_{\text{ans}}$, only $R_{\text{overlap}}$ explicitly targets inter-view entity consistency. However, Table 2 suggests that removing $R_{\text{overlap}}$ yields relatively minor degradation compared to other 2 terms. Does this imply that cross-view reasoning improvements primarily stem from enhanced grounding and answer correctness rather than the explicitly designed cross-view reward? If so, how might the authors further isolate or strengthen the *cross-view component’s contribution*?

---

### Note · Authors · 2025-11-14

I have read and agree with the venue's withdrawal policy on behalf of myself and my co-authors.